# Interpretation of vulnerability and cumulative disadvantage among unaccompanied adolescent migrants in Greece: A qualitative study

Divya Mishra[1,2]*, Paul B. Spiegel[1,3], Vasileia Lucero Digidiki[4], Peter J. Winch[1]

1 Department of International Health, Johns Hopkins Bloomberg School of Public Health, Baltimore, Maryland, United States of America, 2 Geisel School of Medicine, Dartmouth College, Hanover, New Hampshire, United States of America, 3 Center for Humanitarian Health, Johns Hopkins Bloomberg School of Public Health, Baltimore, Maryland, United States of America, 4 François-Xavier Bagnoud Center for Health and Human Rights, Harvard T.H. Chan School of Public Health, Harvard University, Boston, Maryland, United States of America

* divya.mishra@jhu.edu

**Data Availability Statement:** Data cannot be shared publicly because participants are asylum seekers with ongoing legal cases that may be

## Abstract

### Background

In settings of mass displacement, unaccompanied minors (UAMs) are recognized as a vulnerable group and consequently prioritized by relief efforts. This study examines how the interpretation of vulnerability by the national shelter system for male UAMs in Greece shapes their trajectories into adulthood.

### Methods and findings

Between August 2018 and April 2019, key informant interviews were carried out with child protection staff from Greek non-governmental organizations that refer UAMs to specialized children's shelters in Athens to understand how child protection workers interpret vulnerability. In-depth interviews and life history calendars were collected from 44 male migrant youths from Afghanistan, Pakistan, Bangladesh, and Iran who arrived in Greece as UAMs but had since transitioned into adulthood. Analysis of in-depth interviews and life history calendars examined how cumulative disadvantage and engagement with the shelter system altered youths' trajectories into adulthood. Younger adolescents were perceived as more vulnerable and prioritized for shelters over those who were "almost 18" years old. However, a subset of youths who requested shelter at the age of 17 years had experienced prolonged journeys where they spent months or years living on their own in socially isolated environments that excluded them from experiences conducive to adolescent development. The shelter system for UAMs in Greece enabled youths to develop new skills and networks that facilitated integration into society, and transferred them into adult housing when they turned 18 years old so that they could continue developing new skills. Those who were not in shelters by age 18 years could not access adult housing and lost this opportunity. Limitations included possible underrepresentation of homeless youth as well as the inability to capture

affected by data. Full transcripts of in-depth interviews may identify participants even after names and other standard identifying information is removed, and participants' safety may be at significant risk if they are identified. For this reason, making interview transcripts publicly available is contrary to the conditions of ethical approval from the Johns Hopkins School of Public Health Institutional Review Board. Data are available from the Johns Hopkins Center for Humanitarian Health (contact via humanithealth@jhu.edu or mduff4@jhu.edu) for researchers who meet the criteria for access to confidential data.

**Funding:** This study was funded by the Johns Hopkins Center for Qualitative Studies in Health and Medicine's Dissertation Improvement Award (https://www.jhsph.edu/departments/health-behavior-and-society/research-and-centers/center-for-qualitative-studies-in-health-and-medicine/index.html) and the Pulitzer Center on Crisis Reporting (https://pulitzercenter.org). Both awards were given to author DM. Neither awards have grant numbers. The funders had no role in study design, data collection and analysis, decision to publish, or preparation of the manuscript.

**Competing interests:** I have read the journal's policy and the authors of this manuscript have the following competing interests: PBS served as a Guest Editor for the Special Issue on Refugee and Migrant Health.

**Abbreviations:** EKKA, National Center for Social Solidarity; NGO, non-governmental organization; UAM, unaccompanied minor; UNHCR, United Nations High Commissioner for Refugees.

all nationalities of UAMs in Greece, though the 2 most common nationalities, Afghan and Pakistani, were included.

## Conclusions

Due to the way vulnerability was interpreted by the shelter system for UAMs, youths who had the greatest need to learn new skills to facilitate their integration often had the least opportunity to do so. To avoid creating long-lasting disparities between UAMs who are placed in shelters and those who are not, pathways should be developed to allow young adult males to enter accommodation facilities and build skills and networks that facilitate integration. Furthermore, cumulative disadvantages should be taken into account while assessing UAMs' vulnerability. Following UAMs' trajectories into early adulthood was critical for capturing this long-term consequence of the shelter system's interpretation of vulnerability.

## Author summary

### Why was this study done?

- Of the approximately 4,000 unaccompanied minors entering Greece each month, over half are waitlisted for placement in children's shelters due to insufficient capacity.

- Over 90% of unaccompanied minors in Greece, as well as in Europe overall, are older adolescent males who are only eligible for children's shelters for short periods, and little is known about what happens to unaccompanied minors once they age out of eligibility for children's shelters, including those who turn 18 while still on the waitlist.

- A greater understanding of how organizations decide which unaccompanied minors are most vulnerable, and the role that placement in children's shelters plays in unaccompanied youths' life trajectories, can lead to more informed decisions regarding the care of unaccompanied minors.

### What did the researchers do and find?

- In 2018–2019, we (1) interviewed 9 staff members of non-governmental organizations that work with migrant youths in Athens, (2) interviewed 44 male Afghan, Pakistani, Bangladeshi, and Iranian youths who arrived in Greece as unaccompanied minors but had since turned 18, and (3) constructed life history calendars detailing youths' changing circumstances from birth up to the present.

- We found that younger adolescents were prioritized for shelter over 17-year-old minors because younger age was assumed to indicate greater vulnerability, thereby leaving older minors to face additional challenges with survival and integration.

- Placement in children's shelters typically led to placement in adult housing after age 18 and allowed these youths to invest time building new skills and networks that facilitated their integration.

- Unaccompanied minors who turned 18 before being placed in children's shelters often remained homeless or informally housed through early adulthood and continued accumulating disadvantages due to low prioritization of single men for adult accommodations.

- Minors closer to age 18, some of whom were especially vulnerable due to extensive time in socially isolating circumstances of homelessness or child labor in transit countries, were not prioritized for placement in children's shelters because they were considered less vulnerable due to their age.

## What do these findings mean?

- Cumulative disadvantages should be carefully considered when assessing unaccompanied minors' vulnerability because treating younger age as a proxy for vulnerability does not account for how prolonged periods of isolation, homelessness, or child labor in transit countries might shape unaccompanied minors' psychosocial needs in host countries.

- Because placement in children's shelters impacts youths' trajectories even in adulthood, pathways should be developed to allow youths who turn 18 while still waitlisted for shelter to enter adult accommodation facilities and build skills and networks that enable them to integrate into society.

- Without more nuanced indicators of vulnerability, youths who have the greatest need to learn new skills to facilitate their integration may have the least opportunity to do so.

## Introduction

The number of migrant children traveling unaccompanied has been rising globally over the previous decade [1–3]. In Europe, the number of unaccompanied minors (UAMs), or migrants under the age of 18 years traveling without adult family members, spiked from 13,800 in 2013 to 23,300 in 2014, then to an unprecedented 96,000 in 2015. Actual numbers are likely higher, as many UAMs remain undocumented by authorities [4]. In Greece, UAMs made up an estimated 35% of the migrants who arrived in 2015 [5,6]. The restrictive immigration policies that accompanied the EU–Turkey Statement transformed Greece into a de facto buffer zone for migrants [7], where backlogged asylum and family reunification processes kept UAMs in limbo for years [8]. In 2019, Greece hosted an estimated 32,000 migrant children, over 4,000 of whom were unaccompanied or separated [9,10].

UAMs are considered an especially vulnerable subgroup of migrants in humanitarian contexts and are consequently prioritized for aid [11]. Vulnerability, according to the International Organization for Migration, is "the diminished capacity of an individual or group to have their rights respected, or to cope with, resist or recover from exploitation, or abuse" [12]. In Greece, sexual exploitation of UAMs has been an issue of particular concern [13,14]. To protect UAMs, local and international non-governmental organizations (NGOs) in Greece operate shelters for unaccompanied children, which provide psychosocial services of variable kinds and quality to facilitate their integration [15]. However, shelter spaces are limited, and approximately two-thirds of UAMs who requested shelter in 2018 and 2019 were waitlisted

every month [10,16]. Since over 90% of UAMs in Greece, as in Europe overall, are males between 15 and 17 years of age [10,17,18], they are only eligible for shelters for short periods before they turn 18 years old and are categorized as men, at which point they are no longer seen as vulnerable or prioritized for aid [19–21].

Empirical research has found that, contrary to widespread assumptions regarding young men's vulnerability or lack thereof, male gender and traveling unaccompanied are statistically significant indicators of vulnerability to exploitation among minor and adult migrants [12]. No significant difference in indicators of exploitation was found between UAMs and adult males up to the age of 27 years [12]. Substantiating these findings, a Care International study found that unaccompanied single males, adults and minors alike, commonly experienced sexual and economic exploitation in Greece and did not receive the institutional support necessary to exit exploitative circumstances [18]. The continued exploitation of unaccompanied young males even in adulthood may be explained by environmental factors that remain unchanged as youths transition from adolescence into adulthood, or by developmental commonalities among older adolescents and young adult males, as reflected by the fact that the World Health Organization defines adolescence as ages 10–19 years [22].

The trajectories that bring UAMs to Europe encompass various legal statuses, living conditions, and changing motivations [23], with long periods of immobility punctuated by bursts of mobility [23–25]. Many make multiple attempts to arrive at their destinations, while others move on from intended "destination" countries when faced with hardship or unexpected opportunities [26]. Prolonged journeys longer than 3 months are associated with a higher incidence of exploitation, and with traveling unaccompanied and male gender [4,12]. Approximately 25% of UAMs entering Greece experienced journeys lasting 3–6 months, while 13% experienced journeys longer than 6 months [4]. UAMs' experiences during these months remain largely unknown.

Unlike most adolescents, UAMs come of age away from families and communities that can structure their transition to adulthood, through rites of passage [27–29] and experiences that help form adult identities [30,31]. Nonetheless, the different social environments they inhabit [32] influence their physical, psychosocial, and emotional development [33,34]. To understand how these social environments shape UAMs' progression to adulthood, this study draws on life course theory [35,36], which examines transitions that mark adolescents' life trajectories, such as leaving their childhood home, changing legal status, or moving to a new country. Some transitions function as turning points and alter youths' life trajectories.

Life course theory highlights how trajectories of disadvantaged youths are shaped by prior life circumstances [37]. Within life course theory, cumulative disadvantage refers to cycles of reciprocal interactions between an individual's reaction to his or her disadvantage and the environment's response to that reaction, which cause disadvantage to accumulate over years and lead to poor outcomes in adulthood [37]. Adolescents' isolation from society, in particular, has been identified as an important disadvantage with lasting repercussions in adulthood [37,38]. Using life course theory, this study examines UAMs' concurrent developmental and migratory trajectories to understand how their accumulated disadvantages expose them to exploitative circumstances, as well as how the interpretation of vulnerability by Greece's shelter system impacts their trajectories into adulthood.

## Methods

We conducted interviews in Athens from August 2018 to April 2019 with staff from Greek NGOs and with migrant youths. Names of NGOs, child protection workers, and migrant youths were replaced with pseudonyms to protect their identities. Given the scarcity of

information regarding UAMs' trajectories into adulthood, this study was primarily exploratory in nature.

Key informant interviews were carried out with a total of 9 child protection workers at NGO X, NGO Y, and Transitional Shelter Z for young adults (see S1 Text). The 2 NGOs were purposefully selected due to their prominence in child protection efforts in Athens and their close relationships with the Greek government and the United Nations Refugee Agency (United Nations High Commissioner for Refugees [UNHCR]). Transitional Shelter Z was selected because its work with former UAMs offered key insights into unaccompanied migrant youths' transition into adulthood. The number and type of staff interviewed at each organization varied by the size and structure of the organizations. NGO X referred homeless and unstably housed UAMs to shelters and provided holistic case management services until the youths were successfully placed, including legal support, mental health services, accompaniment to medical appointments, and educational activities with peer groups. Two lawyers, a social worker, a psychologist, and a cultural mediator were interviewed at NGO X. NGO Y operated 1 specialized shelter for UAMs, transferred UAMs from the islands to the mainland, provided legal guardians for UAMs, and referred them to other shelters when needed. A social worker and project manager were interviewed at NGO Y. Transitional Shelter Z was staffed by a social worker and psychologist, both of whom were interviewed. It housed 24 young adult males who had arrived in Greece as UAMs. Interviews focused on processes through which youths were placed in shelters.

To understand UAMs' life trajectories, in-depth, semi-structured interviews were conducted with male migrant youths aged 18–21 years who arrived in Greece as UAMs (see S2 Text). To maximize the diversity among participants, initial ethnographic assessment in the form of participant-observation at food distribution sites for the homeless and a youth center for 16- to 21-year-old migrants was used to identify different social groups of migrant youths, largely defined by nationality, history in children's shelters, and current living arrangements. A snowball sampling strategy was used in which seeds were recruited from each social group to both participate in interviews and help identify other eligible youths for recruitment [39]. Potential participants were approached in person or over the phone. Recruitment continued until thematic saturation was reached at 44 interviews [40]. Thematic saturation depended, in turn, on including a sufficient number of study participants from the 2 main source countries for UAMs in Greece, which were Afghanistan and Pakistan [12]; a sufficient number of study participants placed in children's shelters; and a sufficient number of participants who were not placed in children's shelters. Thematic saturation could therefore only be determined as the study proceeded, rather than prospectively [41,42]. Table 1 provides the demographics and relevant characteristics of the 44 youths interviewed. Female UAMs were excluded because they make up less than 10% of UAMs in Europe [17,18] and would require alternative recruitment strategies.

Interviews typically lasted 1 to 2 hours and allowed participants to construct a narrative of the events surrounding their departure from home countries up through the time of the interview. Interviews were conducted by author DM, who had volunteered as an Urdu interpreter on Lesbos (Greek island) in 2016 and had since maintained relationships with Greece's Afghan and Pakistani migrant communities. Interviews were conducted with the support of trained interpreters from migrant communities in Farsi, Dari, or Urdu to target Afghan and Pakistani youths as they represent the most common nationalities among UAMs in Greece [12]. Since the interviewer was female, all interpreters were male in order to limit UAMs' discomfort in discussing sensitive topics. Interviews took place in informal settings selected by participants, such as parks and cafés, where conversations were not heard by individuals other than the participant, interpreter, and DM. All interviews were audio-recorded, then transcribed and

**Table 1. Participant demographics.**

| Characteristic | Number or mean (range) (*n* = 44) | Percent |
|---|---|---|
| Nation of origin | | |
| Afghanistan | 27 | 61.4 |
| Pakistan | 10 | 22.7 |
| Bangladesh | 5 | 11.3 |
| Iran | 2 | 4.6 |
| Age at interview (years) | 18.5 (18–21) | — |
| Age at departure from home country (years) | 15.9 (13–17) | — |
| In children's shelter(s) before age 18 years | | |
| No | 22 | 50.0 |
| Yes | 22 | 50.0 |
| Housing at time of interview | | |
| Homeless | 9 | 20.5 |
| Informally housed | 7 | 13.6 |
| NGO-provided accommodations | 28 | 65.9 |

NGO, non-governmental organization.

translated into English. Due to literacy challenges, transcripts could not be returned to participants for correction. Study aims and procedures were explained, and consent was obtained from all participants at the start of each interview.

For each interviewed migrant youth, a semi-structured life history calendar [43] was constructed to capture the trajectories that participants' lives had taken. The life history calendar took the form of a timeline that identified changes in participants' living situations, how long each living situation lasted, and sources of support that were available in each situation. This method was able to capture how participants' circumstances changed as they moved between countries and within countries, such as from one shelter facility to another. To have comparable data on life history prior to departure for all participants, more structured follow-up interviews were conducted to extend life history calendars to cover the period between birth and departure from the home country, with a focus on (1) place and type of residence during each year of life, (2) sources of economic support in each place of residence, (3) co-inhabitants in each residence, (4) years of schooling, and (5) experience with independent decision-making (see S3 Text). Only 32 of 44 participants could be reached for follow-up interviews. Those who could not be reached had inconsistent phone numbers, left Athens to look for work, or did not respond to voice messages.

Data analysis used a combination of inductive and deductive coding to allow nuanced understandings to emerge from the data while remaining focused on study objectives [44]. Analysis began with the application of broad, predetermined deductive coding categories, including "interpretations of vulnerability" for key informant interviews and "forms of support," "disadvantages," and "access to resources" for in-depth interviews and life history calendars. Inductive coding was then used to identify nuanced themes within those categories, as well as other themes outside those categories that emerged from the data. Coding was carried out by author DM under the supervision of author PJW and managed using QSR International's NVivo 12 software [45]. To understand how placement in children's shelters impacted life trajectories, the life history calendars of youths who were and were not placed in children's shelters were compared, with commonalities and differences documented in memos. Feedback on findings was provided by migrant community members and child protection staff who had participated in the study. This study received approval from the Johns Hopkins Bloomberg

School of Public Health Institutional Review Board (approval number IRB 8788). Oral consent was obtained from participants due to potential risk of identification from written signatures. NGOs were not involved in ethics reviews as they were not involved in recruitment of migrant youth, and staff were only interviewed regarding their professional work.

## Results

This section first presents interpretations of vulnerability identified in key informant interviews, followed by an overview of disadvantages faced by youths in their home countries and during their journeys. Then, 3 case studies are presented to illustrate accumulating disadvantages over UAMs' lifetimes and the pivotal role of children's shelters in shaping life trajectories.

### Interpretations of vulnerability

To place homeless or informally housed UAMs in children's shelters, child protection organizations sent standardized referral forms on their behalf to the Greek National Center for Social Solidarity (EKKA). Securing placement typically took several months, unless the UAM in question was considered exceptionally vulnerable, in which case placement was often arranged within a month. Analysis of key informant interview transcripts revealed 3 major themes regarding how interpretations of vulnerability influenced the prioritization of some UAMs over others for placement in children's shelters. These were (1) differences in how field staff and EKKA perceived a given UAM's vulnerability, (2) variability in the reporting of vulnerabilities, and (3) the importance of age.

**Differing perceptions of vulnerability.** Vulnerability was interpreted differently by field staff who met UAMs in person and EKKA, which primarily learned about them from referral forms, as explained by social worker R and psychologist M from NGO X.

> R: EKKA does not have the opportunity to meet each unaccompanied minor for whom they receive referral for shelters. So that means they don't. . .exactly understand the situation of the minor.

> M: If EKKA [is looking at] 5 cases, and they're all from Pakistan, 16 years old, maybe hosted by some friends now and then, [and] they have no papers—they have the same criteria. How will [EKKA] prioritize? Of course, if they see someone 15, or 14, they prioritize them. . .But sometimes, we see that someone 15 years old feels safe where he is hosted. We can also see someone who is 16 or 16 and a half who doesn't give the impression that he feels safe. . .If the child is dirty, [or] if he is totally homeless [and] it seems as if the child has not taken a bath in 10 days. . .[Or] if he has not had food to eat for 5 days. . .[Then] we decide to prioritize his case.

**Variability in reporting vulnerability.** Social Worker R explained that it was possible to emphasize aspects of vulnerability other than age. However, the reporting of these other aspects was dependent on the staff member who completed the referral and was thus highly variable.

> R: We have a specific form from EKKA. . .The end of the template [has space for] the social history. . .It is in the [case] worker's role to understand what are the vulnerabilities of each minor. . .so that they can write them down in the social history, and EKKA can be informed about why that child is more vulnerable than the other. . .[But] it depends on the case worker.

M: [The case worker] may write 1 or 2 sentences, that [the UAM] is homeless, that he is distressed. [This] vague, general information doesn't help the boy.

**The importance of age.** Though key informants reported that descriptions of factors of vulnerability other than age can counter the assumption that the youngest UAMs are the most vulnerable, their ability to do so seemed to diminish as UAMs approached adulthood. Psychologist M highlighted the challenges faced by homeless and informally housed UAMs as they approached their 18th birthday:

M: The problem is, there is a prioritization of the younger ages. . .If they are almost 18, it is a very gray zone, because they are neither minor—let's say, not very, very vulnerable, though of course, it is not just about their age—but they are not adults yet. We try to help them, but it is very difficult for an almost 18 year old to enter a shelter.

Concurring with Psychologist M, Social Worker V of NGO Y stated:

V: You can understand that, someone who is 16 years old can be easily prioritized to be placed in the facility. Someone who is 2 months until 18 would not be that eligible. I mean, they are eligible under what the law says. But they will not be prioritized.

If youths turned 18 years old before they could be placed in a shelter, they were no longer considered vulnerable and often remained homeless or informally housed in adulthood, as explained by Psychologist M from NGO X.

M: More than 18, and [the youth] is considered a single man. . .Single men are not very prioritized for accommodations, whether it is a shelter, even if it is a camp, [or] it is the apartments [provided] by UNHCR. . .So, it is very difficult when [UAMs] are almost 18.

Supporting M's comments, the social worker at Transitional Shelter Z revealed that 16 of the 24 young adults who lived there had been transferred from children's shelters. The remaining 8 had been transferred from camps, suggesting that it was unlikely for youths who were not placed in any kind of children's shelter as minors to receive accommodations in adulthood.

## Cumulative disadvantages

**Disadvantages prior to departure.** The disadvantages that youths faced in their home countries shaped the experiences they had along their journeys. UAMs whose families were struggling financially often could not pay a smuggler to take the migrant youths all the way to Greece. These youths spent months or years in Iran or Turkey, homeless or working in exploitative conditions. Youths whose families had extensive connections with diasporic communities were often able to seek help from a relative or family acquaintance even though they were unaccompanied. Table 2 describes how disadvantages that youths experienced prior to their departure shaped future experiences, often leading to further disadvantages. This table is not intended to provide an exhaustive list, but rather to illustrate the impact that early disadvantages can have.

**Disadvantages as an undocumented child migrant.** Once youths left their home countries, their experiences fell into 2 broad categories: being transported by smugglers and living alone. The experience of being transported by smugglers was often dangerous and traumatic, as described by Fayaz (Afghan, 18 years old, arrived on Lesbos at age 14).

Fayaz: I started my journey from Nimroz [Afghanistan]. . .Along the route, there were mountains, [and] a desert. Sometimes, we walked on foot for 1 whole day and night, for 24 hours. Sometimes, [they] took us in a car, up to 12 or 16 people in a car the size of a normal taxi. They even put people in the trunk.

My worst memories are of the car . . .They mistreated us, and we couldn't do anything because it wasn't our country. The agents [smugglers] harassed the Pakistanis a lot, violently. They harassed Afghans, too, yanking our hair, and things like that. For someone who hasn't seen all this, he becomes mentally unhinged when he sees it for the first time.

While they were being transported, youths were not necessarily on the move every day. When there was high police activity, or if weather conditions were harsh, migrants were kept waiting for weeks, even months, in smugglers' safe houses, called *musafer khanas* in Urdu and Dari/Farsi. However, the intention of further travel was always present.

There were few independent decisions that youths needed to make—or could make—while being transported by smugglers. The food they ate, the amount of water they drank, where they stayed, and how they traveled were all determined by smugglers and their associates. The following excerpt from Mohammad (Bangladeshi, 19 years old, arrived on Lesbos at age 17) demonstrates the lack of autonomy youths experienced when traveling with smugglers.

Mohammad: One day, I tried to tell [the agent] that I had a fever and couldn't walk. It was cold and raining really hard, and we had to walk outside. When I told the agent, he started to beat me. I was like, "Why are you doing this? I have a fever, and you are beating me?" He said, "You will have to walk. If you stay here, the police will catch you. If the police catch you, and they ask you how you got here, you will tell them about me. . .I'm not going to get caught for you. So you walk. If you die, I will toss your body to the side of the road."

In contrast to when they were being transported, once youths were taken to an agreed upon destination and accounts with smugglers were settled, they were left on their own. This was many UAMs' first experience making decisions without adult supervision. Table 3 gives an overview of the types of circumstances that migrant youths found themselves in while living on their own.

A common disadvantage across these diverse circumstances was the fact that UAMs were socially isolated: They were without a peer group or supportive community and were excluded from activities that would help build networks or skills that they could benefit from in adulthood. UAMs whose families could afford to pay smugglers to take them all the way to Greece often did not experience living alone at all. Consequently, they were often spared long periods of social isolation.

## Case studies

This section follows the experiences of 3 youths as case studies to illustrate how the accumulation of disadvantages shapes migrant youths' trajectories and how children's shelters can alter them. These young men are Gauhar (Afghan, 19 years old, arrived in Greece via land route at age 17), Hafez (Iranian, 18 years old, arrived on Samos [Greek island] at age 16), and Bilal (Pakistani, 19 years old, arrived in Greece via land route at age 17). Table 4 presents their experiences prior to leaving their home countries.

As described in Table 4, Gauhar and Hafez experienced many disadvantages prior to their departure from their home countries. Gauhar's family was displaced from Afghanistan to Pakistan, where he was born. They did not own property, and Gauhar never went to school.

**Table 2. Disadvantages prior to departure.**

| Disadvantage | Explanation | Illustrative quote |
|---|---|---|
| History of displacement | Youths who were displaced with their families prior to traveling unaccompanied often grew up in circumstances where they had limited rights, and their families had few assets. | "I am Afghan, but I was born in Iran. . .Afghan refugees are not allowed to go to school in Iran so I came to Turkey to study. But, I couldn't study in Turkey because I had to work to make a living." —Jamal, Afghan, 19 years old, arrived on Lesbos at age 17 |
| Death or disappearance of parent(s) | Youths who lost one or both parents as children were susceptible to neglect, poverty, limited educational opportunities, and child labor. The death of fathers in particular caused economic hardship. | "When my father passed away, we had no breadwinner in the family to take care of us in Kabul. My aunts and uncles were in Pakistan &so we decided to move there. . .I had to drop out of school, because I had studied up to the eighth grade in Afghanistan. . .If I wanted to study in Pakistan, I would have to start from the first grade." —Mahdi, Afghan, 19 years old, arrived on Lesbos at age 17 |
| Lack of education | Youths who had limited access to education usually could not communicate in English when they arrived in Greece. Some were not literate in any language. Youths who did not spend time in school also had less experience interacting with adults in institutionalized settings. | "When I came to Lesbos, they taught us how to read and write [English], and I learned a little bit. . .I cannot read or write in my first language. If someone messages me [in Dari] on Facebook, I can't write back." —Masood, Afghan, 19 years old, arrived on Lesbos at age 17 |
| Inability to pay smuggler | Youths whose families could only afford to pay a smuggler to take them as far as Iran or Turkey, instead of all the way to Greece, often spent long periods of time homeless or working undocumented before they reached Greece. | "We had only arranged with the smuggler to take me as far as Turkey. . .My mother paid. . .She sold her jewelry. . .and then I stayed and worked. . .I met some other Afghans, and I found [factory] work through them." —Fayaz, Afghan, 18 years old, arrived on Lesbos at age 14 |
| Lack of contacts in diaspora communities | Youths whose families had contacts within diaspora communities sought help from these contacts to avoid homelessness and to access information and resources when they were unaccompanied. | "I didn't have anywhere decent to live. . .I didn't have any relatives in Greece. . .Then I called home and asked if there was anyone from our village here. . .In 2, 3 days, I found someone. . .I went and lived with him for a month." —Hasib, Bangladeshi, 19 years old, arrived on Lesbos at age 17 |

Gauhar's father became too ill to work, at which point Gauhar, aged 10, started working to support the family. Hafez lost both of his parents by the time he was 9 years old, at which point he dropped out of school. He lived with relatives and family friends for short periods of time, and by age 12 had started working in construction. At age 14, he rented an apartment with his little brother. Both Gauhar and Hafez could afford to pay a smuggler to take them only as far as Turkey.

**Table 3. Circumstances experienced while living alone.**

| Circumstance | Explanation | Illustrative quote |
|---|---|---|
| Homeless | While they were homeless, youths had little to no social support. They could not participate in activities or build relationships that might prepare them to be self-sufficient adults, and their desperation to meet basic needs, like food and shelter, left them vulnerable to exploitation. | "When I came to Turkey, I slept on the streets. . .After a month and a half, a Turkish man asked me why I sleep outdoors. I told him I didn't have a place to stay and he took me to his house. . .Two days later, he asked me to work with him on a construction site. I worked with him, but he didn't mention anything about paying me." —Asgar, Afghan, 18 years old, arrived on Lesbos at age 16 |
| Working | Youths who worked while living abroad typically worked in exploitative industries where they were vulnerable to injuries, paid very little, and denied opportunities for personal growth and development as adolescents. Some were loosely supervised by relatives, while others were on their own. | "I lived in Turkey for about a year. . .It was very difficult work. I worked 13 or 14 hours a day in a factory that made and packaged speakers. . .I was just so tired afterwards. I really like football, but I could only watch it on TV, I couldn't play. I was fed up." —Fayaz, Afghan, 18 years old, arrived on Lesbos at age 14 |
| In the care of relatives | Even in the care of relatives they trusted, youths' participation in host societies was limited due to their undocumented status or differences in race, religion, or language. They remained isolated from their peers. | "I was doing well in Iran. . .My uncle took care of me. . .I didn't meet the people of Iran, because I am Sunni and they are Shia. My uncle said, stay home, but if you want to go out. . .don't talk to [Iranians], and don't pray outside. . .If they see you pray [like a Sunni], they might attack you." —Mohammad, Bangladeshi, 19 years old, arrived on Lesbos at age 17 |
| Passing time | A minority of youths were able to request enough money from their families back home, through agencies like Western Union or MoneyGram, and did not need to work while in Iran or Turkey. However, these youths were still undocumented and unable to participate in or integrate into host societies. | "There was nothing for me to do [in Turkey]. I wasn't in a good place. . .I just wandered about, from place to place, to internet cafés. . .I used to call home for money whenever I needed anything." —Tariq, Pakistani, 19 years old, arrived in mainland Greece via boat from Turkey at age 17 |

Relative to his peers, Bilal had several advantages prior to his departure (see Table 4). Though his parents could not afford to send him to university in Pakistan, they owned property and could afford to pay a smuggler to take Bilal directly to Europe. Bilal had also finished high school before he left and was fluent in English. Table 5 below presents the youths' experiences while traveling unaccompanied.

Bilal (see Table 5) was transported by smugglers for the entire duration of his 1-month journey from Pakistan to Greece. Furthermore, his father gave him additional cash to pay smugglers for better treatment. In contrast, Gauhar and Hafez (Table 5) had only paid to be taken as far as Turkey, where they began living alone. Gauhar was in Turkey for 4 years, during which time he was homeless and then paid rent to stay in someone's basement. He collected cardboard scraps to earn money, most of which he spent on rent. It was only when he felt threatened by a local trafficking gang that he paid a smuggler to take him to Greece. Hafez spent 6 months working in a clothing factory in Turkey before he received an unexpected opportunity to go to Greece. Table 6 presents the youths' experiences in Greece.

**Table 4. Comparative case studies: Home countries.**

| Case | Trajectory |
|---|---|
| Gauhar (Afghan, 19 years old, arrived via land route at age 17) | Gauhar was born in Peshawar, Pakistan, where his family lived as registered refugees. He lived with his mother, father, and 4 siblings in a dirt house that his father rented from the money he earned as a taxi driver. Gauhar never went to school.<br>When Gauhar was 10 years old, his father was diagnosed with cancer and could no longer work. The family returned to Afghanistan and rented a house in rural Langarhar, where his older brother began working as a rickshaw driver and Gauhar worked as a part-time butcher's assistant to support the family and pay for their father's medical bills. When Gauhar was 13 years old, the local Taliban attempted to recruit him, and his mother told him to leave Afghanistan. His brother paid a smuggler US $1,200 to take him to Turkey. |
| Hafez (Iranian, 18 years old, arrived on Samos at age 16) | Hafez was born in Mashhad, Iran. His father passed away when he was 2 years old, at which time his mother moved him and his younger brother to an apartment in the town of Gonbad, where she worked as a teacher. When Hafez was 9 years old, his mother, too, passed away. Hafez stopped attending school after his mother died.<br>Hafez and his brother spent 3 years living with their aunt, who Hafez found intolerably harsh. When he was 12 and his brother 11, they left their aunt's house and were briefly homeless. However, a man who knew their mother offered to let them stay in a room above his garage in exchange for their assistance at his mechanic's shop. Hafez worked at a construction site while his brother assisted the mechanic. When Hafez was 14 years old, he and his brother rented an apartment of their own. An elderly Christian woman who lived nearby helped the boys frequently —"She became like my mother," Hafez said—and Hafez decided to convert to Christianity, which constituted a crime in Iran. When some relatives threatened to report him to authorities, an aunt paid a smuggler to take him to Turkey, where he could avoid persecution. At the time, he was 16 years old. |
| Bilal (Pakistani, 19 years old, arrived via land route at age 17) | Bilal was born in Peshawar, Pakistan, into a large joint-family household. His mother came from an educated family of lawyers, whereas his father's family was implicated in local gang violence. When Bilal was 14 years old, his parents sent him to an English medium boarding school in a different town, away from the violence that the family was embroiled in. At the age of 17, Bilal's parents told him they couldn't afford to send him to university. After a family discussion, Bilal and his parents decided to send him to Europe, in hopes that he could build a life, and maybe even continue his education in a place safe from violence. With money borrowed from relatives, Bilal's family paid a smuggler to take him to Italy. |

**Table 5. Comparative case studies: Journeys.**

| Case | Trajectory |
|---|---|
| Gauhar (Afghan, 19 years old, arrived via land route at age 17) | When Gauhar arrived in Turkey, he was undocumented, had nowhere to go, and slept in a park. While he was homeless, he saw other migrants collecting cardboard boxes from the streets and trash bins. He asked these other migrants and discovered that he could get paid for collecting cardboard. He began collecting and selling cardboard as well, and earned about 800 lira (US$130) per month. Not long after he started doing this work, a Kurdish man saw him sleeping in the park and offered him a basement room for 500 lira per month. For 4 years, Gauhar spent his days collecting cardboard and slept in the Kurdish man's basement. When Gauhar was 17 years of age, he got into a knife fight with a local trafficking gang when they wanted to sell a homeless woman he was close to. From that point on, he continued to face threats of violence from the gang and decided to leave Turkey in search of safety, using his savings to pay a smuggler to take him to Greece. |
| Hafez (Iranian, 18 years old, arrived on Samos at age 16) | When he crossed the border into Turkey, Hafez boarded a bus to Istanbul, where he went to a neighborhood nicknamed Iranian Street to look for a job. He found work in a factory making jeans and paid rent to stay in an apartment with 4 other Iranians. He had been working in the factory for 6 months when his smuggler contacted him saying that his aunt had paid him to take Hafez to Greece. Hafez was still 16 years old when he left for Greece. |
| Bilal (Pakistani, 19 years old, arrived via land route at age 17) | Bilal recalled his journey to Europe as terrifying. He witnessed Iranian border security shoot at migrants, was made to walk for entire days without water, and was ordered to run through the night by the smugglers. He was slapped by a smuggler once for lagging behind the group. However, he said the journey was often safer than that of his co-travelers. Prior to his departure, Bilal's father had equipped him with US dollars, the dominant currency in Asia's smuggling networks, and instructed him to tip the guides and drivers who transported him to avoid harassment. By tipping preemptively, Bilal believed he was usually able to secure relatively comfortable arrangements, even as he watched other travelers get forced into the trunk of a car by smugglers. It took Bilal approximately 1 month to reach Greece, during which he was continuously transported by smugglers. |

After their arrival in Greece, Hafez and Bilal were eventually, though not immediately, placed in shelters for UAMs (Table 6). Shelters facilitated access to Greek lessons, and both young men were fluent in Greek at the time they were interviewed. Bilal was able to resume his education and had plans to apply for university. Hafez did not pursue higher education but intended to participate in the Greek economy by opening up a fruit stand. Both Hafez and Bilal were transferred to adult accommodations after they turned 18, where they were able to continue learning Greek, attend school, and pursue activities that facilitated their integration into Greek society. Since they lived in NGO-provided housing, they did not have to pay rent. They also received cash assistance from UNHCR to help cover their basic needs.

Despite having requested shelter while he was a minor, Gauhar was not placed in a shelter before he turned 18. Since he was homeless at 18, he could not apply for the cash assistance that most adult asylum seekers received, as the application required a mailing address. Unable to find other work, Gauhar engaged in transactional sex to survive, and considered trying to go to jail in order to sleep indoors.

## Discussion

In key informant interviews, staff from NGO X acknowledged that age was not the only determinant of UAMs' vulnerability, and that other factors, such as whether or not the UAM felt

**Table 6. Comparative case studies: Greece.**

| Case | Trajectory |
|---|---|
| Gauhar (Afghan, 19 years old, arrived via land route at age 17) | Upon arriving in Greece, Gauhar was intercepted by police and detained for 1 month. After he was released, he worked at a construction site for 3 days and made €100. He then bought a train ticket to Athens, where he registered for asylum. In Athens, he requested shelter but turned 18 before he was placed in a shelter. At the time he was interviewed, Gauhar had been homeless in Victoria Square for approximately 8 months. He wasn't able to find a job, and without an address, he could not apply for the monthly €150 cash assistance that most adult refugees received from UNHCR. For approximately 4 of those 8 months, Gauhar reported having sex with Greek men for €10–€20 per customer. He said he was tired of living in Victoria Square, and was considering selling drugs or getting into a fight so that the police would take him to jail, where he would have a roof over his head. |
| Hafez (Iranian, 18 years old, arrived on Samos at age 16) | As soon as Hafez reached Samos, he was taken to a nearby reception center. The reception center did not permit migrants to leave the center for a period of 20 days, at which point they were given their asylum applicant cards and were allowed to move freely on the island, but not to the mainland. Before Hafez's 20 days were up, a group of Iranians invited him to join them as they attempted to stow away on a cargo ship headed for the mainland. Hafez successfully made it to Athens, but was undocumented.<br>Without anywhere to go in Athens, Hafez was taken in by a group of older Iranians in Elefsina Camp in exchange for household work. This arrangement lasted 3 months, during which Hafez was abused and beaten, but kept a low profile due to his undocumented status. Eventually, Hafez was discovered by a social worker who facilitated his asylum application and transferred him to a shelter for unaccompanied minors. At the shelter, Hafez attended Greek classes, and was fluent at the time he was interviewed. When he turned 18, he was transferred from the shelter to an apartment for adults, where he could continue taking language classes without worrying about rent payments. He also received €150 a month in cash assistance from UNHCR. In 2019, he was making plans to open up a small fruit stand in Athens. |
| Bilal (Pakistani, 19 years old, arrived via land route at age 17) | Upon arriving in Greece, Bilal realized he had been duped by his smuggler and would not be taken to Italy after all. He had nowhere to go when he arrived in Athens, and was homeless in Victoria Square for 1 week, informally hosted by other migrants for 1.5 months, and given temporary accommodations by a local NGO for 2 weeks. During that time, Bilal advocated to be placed in a shelter with the assistance of several NGO staff. Four months before he turned 18, Bilal was placed in the children's section of a camp. Six days after his 18th birthday, he was transferred to an apartment for young adults, where he had been living for a year at the time he was interviewed.<br>Since he was placed in the camp, Bilal had taken Greek lessons and repeated the 11th grade in Greece. He was looking for a summer job when he was interviewed, and had plans to repeat the 12th grade in Greece as well, with plans to eventually study engineering in university. As an adult, he received €150 a month in cash assistance from UNHCR. |

NGO, non-governmental organization; UNHCR, United Nations High Commissioner for Refugees.

safe where he was staying and whether he was able to afford food and stay clean, may be even more indicative of vulnerability than age. However, the communication of these other, more subjective factors was dependent on the individual staff members who sent referral forms to EKKA. If these subjective factors were not described convincingly enough, EKKA defaulted to using UAMs' chronological age as a proxy for vulnerability. The ability of other factors to counter the perceived inverse relationship between age and vulnerability diminished after UAMs turned 17, at which point NGO staff described them as "not that eligible" or "neither

minor. . .but not yet adult." If UAMs could not be placed in children's shelters before they turned 18, they often remained homeless or informally housed even as adults.

Neither the association of vulnerability with younger ages nor the more subjective indicators of vulnerability cited by NGO staff took into account how the different experiences that UAMs had before requesting to be placed in a shelter in Greece may have contributed to their vulnerability. Analysis of youths' life trajectories using the lens of cumulative disadvantage [37] found that various disadvantages that youths had in their home countries, such as poverty, prior history of displacement, and lack of education, predisposed them to other disadvantages later in their trajectories, in particular social isolation. Youths from poorer families often could not pay a smuggler to take them all the way to Europe, and therefore these youths spent months or years either homeless or working as child labor, exposed to traumatic or exploitative conditions in socially isolating environments not conducive to adolescent development. Cultural and geographic distance weakened their ties with family back home [2], and their undocumented status, along with cultural and linguistic differences, prevented them from integrating with peer groups in host countries. Living alone in such detrimental, isolated circumstances was a disadvantage that precipitated future disadvantages; the longer UAMs were living alone before arriving in Greece, the longer they were denied opportunities to build skills and networks that would help them become self-sufficient adults and integrate into Greek society. Youths who had fewer disadvantages experienced fewer and shorter periods of social isolation, avoided many exploitative circumstances due to financial support from their families, and had more skills that facilitated integration in Europe.

Once UAMs arrived in Greece, placement in a children's shelter constituted a turning point [36] that had the potential to counter the consequences of prior disadvantages, including social isolation and lack of education. In addition to providing for UAMs' basic needs, shelters also gave them opportunities to network with peers their own age who had similar backgrounds and experiences. In addition, shelters connected youths to activities that could help them build new skills that would facilitate their integration in Greece. These included Greek and English language classes, vocational training, and, in some cases, guidance on applying for higher education. Importantly, when these youths turned 18, they were transferred to adult accommodations where they could continue investing time in personal development and learning to participate in Greek society without worrying about basic needs like shelter.

On the other hand, UAMs who could not be placed in a shelter before they turned 18 were typically not able to access any accommodation facility at all as young adult males, as they were not considered vulnerable. Such young men continued being homeless or relying on underground economies for survival, which contributed to their social marginalization and exposed them to exploitative circumstances, irrespective of whether they were minors or adults.

The potential of children's shelters to counter the consequences of cumulative disadvantage is illustrated by the case studies of Gauhar, Hafez, and Bilal. Like Gauhar, Hafez faced multiple disadvantages in his home country and had a prolonged journey during which he was living on his own in socially isolating circumstances. However, he, like Bilal, was placed in a children's shelter, where he received many more opportunities to integrate into Greek society than Gauhar. At the time they were interviewed, both Hafez and Bilal lived in NGO-provided housing and participated in some kind of skill-building activity. In contrast, Gauhar remained homeless, socially marginalized, and exposed to exploitative circumstances. Turning 18 did not make Gauhar any less vulnerable than he was as a minor. His environment did not change, except that the possibility of receiving aid diminished even further. The continued lack of stable accommodations further added to his cumulative disadvantage, as the lack of an address prevented him from applying for forms of assistance that he would have otherwise been eligible for

as an adult asylum seeker. Gauhar's case illustrates how accumulating disadvantages can diminish youths' ability to protect themselves from abuse and exploitation even in early adulthood.

The use of life course theory and cumulative disadvantage as analytical lenses [36,37] allowed for an understanding of how disadvantages in UAMs' home countries can lead to prolonged journeys [38], and shape the kinds of experiences UAMs have during prolonged journeys. In particular, data regarding UAMs' experiences during prolonged journeys highlight the pervasiveness of social isolation while UAMs are living on their own. Given that prior research shows the duration of social isolation among adolescent males to be associated with difficulty integrating with society in adulthood [37,38], the duration of UAMs' journey may be an important indicator of youths' psychosocial needs and consequent vulnerability. Furthermore, by examining youths' trajectories into early adulthood, this study captured the potential of children's shelters to have long-term impacts on UAMs' trajectories by countering some of the consequences of prior disadvantages, including social isolation. While some longitudinal studies incidentally capture UAMs who age out of services for minors [46,47], most studies focus on these youths only while they remain underage, and are therefore unable to capture long-term effects of interventions.

Limitations of the study included our inability to reach all youths for follow-up interviews. Some youths had already moved out of Athens to find work by the time they were contacted, while others had left Greece altogether. Some could not be reached at all. Due to their typically smaller social networks, it is possible that homeless youth were underrepresented. The study also did not capture all nationalities of UAMs in Greece, including those who come from countries in Africa or the Arab world. However, the 2 most common nationalities among UAMs, Afghan and Pakistani, were well represented. Furthermore, the results of this study may not be applicable to female UAMs, as their experiences along clandestine journeys and with NGOs may differ significantly from those of males.

In order to avoid creating long-lasting disparities between youths who are placed in shelters and those who are not, pathways should be developed to allow young adult males to enter accommodation facilities and build skills and networks that enable them to integrate into Greek society. Furthermore, cumulative disadvantages [37] should be taken into account when assessing UAMs' vulnerability. Structured reporting of certain parameters in the referral forms distributed by EKKA, such as the duration of youths' journeys and histories of education and child labor, may encourage systematic prioritization of the most vulnerable UAMs instead of relying on variable reporting by NGO staff. Keeping in mind the long-term consequences for youth who are not placed in shelters before they turn 18, 17-year-old youths from significantly disadvantaged backgrounds should be prioritized as urgent cases for protection, or targeted by other programs that facilitate integration after they turn 18. Youths from disadvantaged backgrounds may have more unmet needs than those with fewer disadvantages, even if they are older in age. The assumption that, as they get closer to becoming adult males, these youths are less vulnerable and therefore less in need of assistance [19,21,48] may inadvertently increase their exposure to exploitation.

## Supporting information

**S1 COREQ checklist. COREQ reporting checklist for qualitative studies.**
(DOCX)

**S1 Text. Key informant interview guide.**
(DOCX)

**S2 Text. In-depth interview guide.**
(DOCX)

**S3 Text. Life history calendar for follow-up interview guide.**
(DOCX)

## Author Contributions

**Conceptualization:** Divya Mishra, Paul B. Spiegel, Vasileia Lucero Digidiki, Peter J. Winch.

**Data curation:** Divya Mishra.

**Formal analysis:** Divya Mishra.

**Funding acquisition:** Divya Mishra.

**Investigation:** Divya Mishra.

**Methodology:** Divya Mishra, Peter J. Winch.

**Project administration:** Divya Mishra.

**Resources:** Vasileia Lucero Digidiki.

**Supervision:** Paul B. Spiegel, Peter J. Winch.

**Writing – original draft:** Divya Mishra.

**Writing – review & editing:** Divya Mishra, Paul B. Spiegel, Vasileia Lucero Digidiki, Peter J. Winch.

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
