## [Decision Letter · Decision Letter 0]

23 Dec 2019

Dear Dr. Mishra,

Thank you very much for submitting your manuscript "“Neither minor, but not yet adults”: Interpretations of vulnerability and cumulative disadvantage for unaccompanied adolescent migrants in Greece" (PMEDICINE-D-19-03759) for consideration at PLOS Medicine. 

[LINK]

In light of these reviews, I am afraid that we will not be able to accept the manuscript for publication in the journal in its current form, but we would like to consider a revised version that addresses the reviewers' and editors' comments. Obviously we cannot make any decision about publication until we have seen the revised manuscript and your response, and we plan to seek re-review by one or more of the reviewers. 

We expect to receive your revised manuscript by Jan 06 2020 11:59PM. Please email us (plosmedicine@plos.org) if you have any questions or concerns.

We look forward to receiving your revised manuscript. 

Sincerely,

Clare Stone, PhD

Managing Editor 

PLOS Medicine

plosmedicine.org

Please note Ref 3's comments are within the attached PDF - do let us know if you have any difficulty accessing. 

Please revise your title according to PLOS Medicine's style. Your title must be nondeclarative and not a question. It should begin with main concept if possible. "Effect of" should be used only if causality can be inferred, i.e., for an RCT. Please place the study design ("A randomized controlled trial," "A retrospective study," "A modelling study," etc.) in the subtitle (ie, after a colon).

Please ensure all questionnaires in relation to interview questioning are provided either as citation of published interviews or submitted as Supp Files. 

Abstract: please specify dates with months as well as years; please provide some summary demographic information; Please use ‘youths’ instead of ‘youth’.

We appreciate the data cannot be made available, for the reasons you outline. You provide 2 email addresses for researchers who may be eligible to access data via contacting these persons. Note Paul Spiegel cannot be a point of contact as he is an author. Please delete. The remaining contact is fine if it isn’t an author. 

Refs in main text – please use square instead of round brackets. 

Line 107 – please define “holistic case management services”

Did your study have a prospective protocol or analysis plan? Please state this (either way) early in the Methods section.

Please provide table in the main text of the demographic information re the boys included in the study: country of origin, age when arrived in Greece and other information. 

c) In either case, changes in the analysis—including those made in response to peer review comments—should be identified as such in the Methods section of the paper, with rationale.

Please provide a reporting checklist – perhaps COREQ https://www.equator-network.org/reporting-guidelines-study-design/qualitative-research/?post_type=eq_guidelines&eq_guideli

Comments from the reviewers:

Reviewer #1: This is a statistical review of manuscript PMEDICINE-D-19-03759. This manuscript describes the results of interviews and life history calendars of 44 male migrant youth who arrived in Greece as unaccompanied minors. 

The research methods are mostly qualitative. Life-course theory belongs to the social sciences. Nonetheless, from a statistical perspective, some clarifications would improve the manuscript. My comments are listed below.

Methods line 114: 9 staff members from three organizations were interviewed. I suggest that you clarify how many were interviewed by organization. Is it one staff member for two of the organisations, and 7 from the third organisation, or it is more balanced? 

Methods line 117: could you please explain your sample size? Why did you chose 44 youth? Is it due to logistical/feasibility reasons, or did you "calculate" that with 44 youth your interviews would be sufficiently "representative"?

Methods line 124: please could you explain what snowball sampling is for the reader? In particular, could you provide the total pool of potential UAMs in each subgroup, and how many you interviewed per subgroup? 

Introduction line 57: Greece hosts *an* estimated 32,000…

Reviewer #2: General Comment:

This is a very interesting analytical piece on the vulnerabilities and cumulative disadvantage for unaccompanied adolescent migrants (UAMs) in Greece. It is highly relevant to the ongoing debate on the need to facilitate the social protection of refugees and migrants, particularly children and adolescents. However, the paper would benefit from a more detailed description and in-depth analysis of potential factors that have led to the vulnerability of male UAMs. It would also benefit from a comparative approach to assessing the experiences of interviewed UAMs.

1. The research question is an important one to the community of researchers in this general area

Studying the effect of displacement on asylum seekers is important in the context of population displacement due to armed conflict. Among these population, the most vulnerable subgroup is children, especially those with no family or companionship. The study is worth publishing as it assesses vulnerable groups who lack social support.

2. The results provide a substantial advance over existing knowledge, with clear implications for patient care, public policy, or clinical research agendas

The results of this study can be used to guide public policies and international organizations proving relief and aid for disadvantaged un-accompanies minors and children, as well as adults in host countries.

3. Published together with an Author Summary written for general readers, the article is of interest to clinicians and policymakers who are not specialists in this topic

This paper discussed an important topic that would be of interest to PLOS Medicine readership. I did not have access to the Author Summary written for general readers.

4. What are the main claims of the paper and how significant are they for the discipline?

This study is important since it reports on disadvantage subgroups among a vulnerable population. The study has shown that children who are not placed in shelters become more disadvantaged. It also shows that age is not the only factor to be considered when assessing vulnerability, cumulative disadvantages such as current living conditions, work, and other factor are just as important. The findings of this paper can help guide international organization and other aid programs in improving the relief efforts and maximizing efficiency of the aids targeting this population.

Operationalizing vulnerability in the paper is based only on the views of some representatives from three organizations. It would be helpful to provide more details on this concept (e.g., a lawyer was interviewed, so what are the legal issues concerning operationalizing vulnerability).

It would also be helpful to provide more insights on the shortcomings of how the organizations operationalize vulnerability and explicitly list the main themes that emerged (e.g., focus on age, etc.). The themes could be organized din subheadings to guide the readers.

5. Are the claims properly placed in the context of the previous literature? Have the authors treated the literature fairly?

Although the authors refer to the life course theory in the introduction and the objectives, further details on how the theory was used are missing. Links to the theory should be provided.

The literature review can benefit from the addition of a number of recent articles addressing the topic. A number of recent articles assess the experiences of displaced unaccompanied minor refugees in Europe have been missed. 

Examples of recent studies on unaccompanied minors: 

a. Ferrara, P.; Corsello, G.; Sbordone, A.; Nigri, L.; Caporale, O.; Ehrich, J.; Pettoello-Mantovani, M. The "Invisible Children": Uncertain Future of Unaccompanied Minor Migrants in Europe. The Journal of Pediatrics 2016, 169, 332-333.e331, doi:https://doi.org/10.1016/j.jpeds.2015.10.060.

b. Pradella, F.; Pinchi, V.; Focardi, M.; Grifoni, R.; Palandri, M.; Norelli, G.A. The age estimation practice related to illegal unaccompanied minors immigration in Italy. The Journal of forensic odonto-stomatology 2017, 35, 141-148.

c. Abbing, H.D. Age determination of unaccompanied asylum seeking minors in the European Union: a health law perspective. European journal of health law 2011, 18, 11-25, doi:10.1163/157180911x546101.

6. Do the data and analyses fully support the claims? If not, what other evidence is required?

The analysis of the qualitative data is supportive of the claims presented in this paper.

However, the way the results were presented could be improved. The authors say that: "the following sections follow the experiences of three youths whose experiences were representative of the 44 study participants " .

I suggest to omit the word 'representative' since this is a qualitative study that mainly narrates the experience of three male youth.

One thing that would make the paper stronger is a bit more elaboration on the issue of determination of vulnerability. It seems to me that there is something important to say about the process through which the youth are evaluated, referred and then characterized as more or less vulnerable. Reading the first subsection of the "Results" section gives the impression that part of the problem could be in how some NGO staff represent the children's cases to the EKKA, for example "[The case worker] may write one or two sentences, that [the UAM] is homeless, that he is distressed. [This] vague, general information doesn't help the boy." (page 9, lines 179-80). Or that the problem may be an issue in the procedure, the forms that the NGOs send and the kind of information requested in these forms. Are these forms prepared by the EKKA? Do they need to add more informative questions or sections where the NGO workers can include a bit of the history of the children and their current work/living conditions? 

What are the recommendations of the authors to improve how vulnerability is operationalized? How can the cumulative vulnerability be taken into account when assessing the needs of the UAMs?

7. PLOS Medicine encourages authors to publish detailed methods as supporting information online. Do any particular methods used in the manuscript warrant such publication? If a protocol is already provided, for example for a randomized controlled trial, are there any important deviations from it? If so, have the authors explained adequately why the deviations occurred?

The interview guides were not provided by the authors.

8. Is this paper outstanding in its discipline? If yes, what makes it outstanding? If not, why not?

This paper helps guide the aid efforts of international and local organization to improve the conditions of vulnerable and unaccompanied minors.

9. Are details of the methodology sufficient to allow the experiments to be reproduced?

It would be helpful to clarify the rationale for selecting only three out of 44 male UAMs in order to reflect in detail on their experiences since a larger and more varied group was not selection.

In the first paragraph of the methods (line 103-116), the authors did not specify how and why the 2 NGOs and Shelter were identified and selected for the study. Also since the interviews were not in English, did they translate them? This was not mentioned. In addition to interview guides were not provided.

10. Is the manuscript well organized and written clearly enough to be accessible to non-specialists?

The author in the background and methods section explained most of the terms and concepts used in this paper, so that readers can follow and understand the findings of this paper. However, It would be helpful to refer to a definition of 'cumulative disadvantage' as was the case for vulnerability.

As for the organization, the overall paper is well organized, however the results section could have been better presented, especially the tables. The flow of the information in the results can be improved by separating the sections on the general results (of the 44 youths) and those of the three youths. 

Spelling and grammar mistakes were also identified in this paper.

11. Other comments

1)Title: 

Please revise the title. The use of "neither" is followed by "nor". For example: Neither minors nor adults. 

Make sure to use either singular or plural: minors/adults or minor/adult.

"Interpretation of vulnerability and cumulative disadvantage":

This sentence reads as interpretation of vulnerability and interpretation of cumulative disadvantage. The way the short title is worded is better. Another option is to reverse the title:

Cumulative disadvantage and interpretation of vulnerability for unaccompanied adolescent migrants in Greece.

2) ABSTRACT:

"Operationalization of vulnerability":

Operationalization is different than interpretation. The first follows a specific, agreed-upon set of criteria or rules whereas the second is a matter of personal evaluation and assessment. The two terms should not be used interchangeably. This is important in order to clearly frame and present the issue or problem presented in this paper.

"Due to the way vulnerability was interpreted and operationalized by the shelter system for UAMs---":

This sentence needs clarification. What is the consequence of the shelter system's interpretation of vulnerability?

Is it:

a) that youth who requested shelter... had the greatest need to learn new skills to facilitate their integration?

Or

b) that youth who had the had the greatest need to learn new skills to facilitate their integration often had the least opportunity to do so?

3) Other potential limitation is restricting the sample to male UAMs. This was not mentioned in the limitations, only in the introduction of the manuscript.

4) In the Background section, the author could talk more about the type of relief efforts targeted for this vulnerable population in general.

5) Line 103: replace "from" by "with"

6) Line 212: replace "They family" by "Their families" 

7) In table 2, Under Bilal: replace "whereas his father's side of the was implicated" by "whereas his father's side was implicated"

8) Table 3, in the row on "In the care of relatives": replace: "differences" by "differences in"

9) Table 4, in the first column, in the last paragraph: replace "he savings" by "his savings" 

10) Table 4, in the third column, in the last paragraph: replace "to arrive in Greece" by "to arrive to Greece"

11) Line 281: replace "he could not to apply" by "he could not apply"

12) Line 281: what do you mean by "sold sex"? Please consider rephrasing

13) Table 5, first column: "he not been able to find a job" by "he hasn't been able to find a job"

14) Table 5, second column, last paragraph: replace "transferred to a shelter" by "transferred him to a shelter" 

15) In table 5, there is a sentence |See chapter 2 for details" - Where is chapter 2? 

16) Line 319: replace "this is illustrated the trajectory" by "this is illustrated by the trajectory"

17) Line 335: replace "which exposed them exploitive circumstances" by "which exposed them to exploitive circumstances"

18) Table 1 and 2 has the same title (maybe the author could merge them together). 

19) I suggest putting table 4 before table 3 (and their corresponding paragraphs), this way it is more in line with the timeline: first their experience in transit to Turkey or Iran and then their experience living alone.

20) The titles of all the table could be improved. For example: comparative case studies, part 3 - what does this mean? Where is part 1 and 2?

21) Also the tables could be better presented for organization and visual purposes. 

22) In line 198 to 200, why are tables 1 and 3 not explained (as what to do they present) as the rest of the tables?

[LINK]

---

## [Decision Letter · Decision Letter 1]

12 Feb 2020

Dear Dr. Mishra,

Thank you very much for re-submitting your manuscript "Interpretation of vulnerability and cumulative disadvantage among unaccompanied adolescent migrants in Greece" (PMEDICINE-D-19-03759R1) for review by PLOS Medicine.

I have discussed the paper with my colleagues and the academic editor and it was also seen again by one of the original reviewers. I am pleased to say that provided the remaining editorial and production issues are dealt with we are planning to accept the paper for publication in the journal.

[LINK]

We look forward to receiving the revised manuscript by Feb 19 2020 11:59PM. 

Sincerely,

Clare Stone, PhD

Managing Editor 

PLOS Medicine

plosmedicine.org

Requests from Editors:

Title –Thank you for making the previous change - we still need to add a study descriptor. I suggest:

Interpretation of vulnerability and cumulative disadvantage among unaccompanied adolescent migrants in Greece: a qualitative study 

In the abstract, please add brief demographic details of interviewees and countries of origin and some brief information about the field workers. Finally in the abstract, - the limitations section in the abstract could quote one additional limitation, e.g. the question of selection bias, perhaps

Author Summary – thank you for providing, it’s very clear. Can you please however add into the main text immediately after the abstract.

Paul's status as Guest Editor for the issue needs to be added to the CoI statement, please

Please complete and add the COREQ checklist as a supp file, using sections and paragraphs (not page numbers as these alter on formatting) 

(https://www.equator-network.org/reporting-guidelines/coreq/)

Comments from Reviewers:

Reviewer #3: Thank you for the opportunity to re-review this very relevant article on UAMs and vulnerabilities in Greece. Many thanks to the authors for addressing all of the reviewer comments. I am happy to accept this article as my main concerns have been addressed:

1. The methods section is now clearer, including recruitment process, and has been revised according to the reviewer comments. 

2. Results section - are now better organised into sub-sections, with further discussion of case studies in a subsection and further overview of results.

3. The discussion section has been reorganised for greater clarity, to follow the themes of the results section and there are now clearer tables to outline the results.

4. Interpretation of 'vulnerability' has been further defined and expanded, to make this clearer throughout, as has the life course approach.

This article will not only contribute to the overall academic literature on UAMs, but also to the current very topical debate at operational level about the gap in provision of care for UAMS, and particularly the use of the 'vulnerabilities' criteria.

[LINK]

---

## [Editor Report · Decision Letter 2]

9 Mar 2020

Dear Dr. Mishra, 

On behalf of my colleagues and the academic editor, Dr. Kolitha Wickramage, I am delighted to inform you that your manuscript entitled "Interpretation of vulnerability and cumulative disadvantage among unaccompanied adolescent migrants in Greece: A qualitative study" (PMEDICINE-D-19-03759R2) has been accepted for publication in PLOS Medicine. 

PRODUCTION PROCESS

PRESS

PROFILE INFORMATION

Thank you again for submitting the manuscript to PLOS Medicine. We look forward to publishing it. 

Best wishes, 

Clare Stone, PhD

Managing Editor 

PLOS Medicine

plosmedicine.org